# Advantages and Requirements in Time Resolving Tracking for Astroparticle Experiments in Space

Matteo Duranti [1,*], Valerio Vagelli [1,2,*], Giovanni Ambrosi [1], Mattia Barbanera [1,3], Bruna Bertucci [1,4], Enrico Catanzani [1,4], Federico Donnini [1], Francesco Faldi [4], Valerio Formato [5], Maura Graziani [1,4], Maria Ionica [1], Lucio Moriconi [4], Alberto Oliva [6], Andrea Serpolla [4], Gianluigi Silvestre [1,4] and Luca Tosti [1]

[1] Istituto Nazionale Fisica Nucleare (INFN)—Sez. Perugia, 06100 Perugia, Italy; giovanni.ambrosi@infn.it (G.A.); mattia.barbanera@infn.it (M.B.); federico.donnini@pg.infn.it (F.D.); maria.ionica@pg.infn.it (M.I.); luca.tosti@pg.infn.it (L.T.)

[2] Agenzia Spaziale Italiana (ASI), 00133 Roma, Italy

[3] Dipartimento di Ingegneria dell'informazione, Università di Pisa, 56122 Pisa, Italy

[4] Dipartimento di Fisica e Geologia, Università Degli Studi di Perugia, 06100 Perugia, Italy; bruna.bertucci@unipg.it (B.B.); enrico.catanzani@studenti.unipg.it (E.C.); francesco.faldi@studenti.unipg.it (F.F.); maura.graziani@unipg.it (M.G.); lucio.moriconi@studenti.unipg.it (L.M.); andrea.serpolla@studenti.unipg.it (A.S.); gianluigi.silvestre@studenti.unipg.it (G.S.)

[5] Istituto Nazionale Fisica Nucleare (INFN)—Sez. Roma Tor Vergata, 00133 Roma, Italy; valerio.formato@roma2.infn.it

[6] Istituto Nazionale Fisica Nucleare (INFN)—Sez. Bologna, 40126 Bologna, Italy; alberto.oliva@bo.infn.it

[*] Correspondence: matteo.duranti@infn.it (M.D.); valerio.vagelli@asi.it (V.V.)

**Abstract:** A large-area, solid-state detector with single-hit precision timing measurement will enable several breakthrough experimental advances for the direct measurement of particles in space. Silicon microstrip detectors are the most promising candidate technology to instrument the large areas of the next-generation astroparticle space borne detectors that could meet the limitations on power consumption required by operations in space. We overview the novel experimental opportunities that could be enabled by the introduction of the timing measurement, concurrent with the accurate spatial and charge measurement, in Silicon microstrip tracking detectors, and we discuss the technological solutions and their readiness to enable the operations of large-area Silicon microstrip timing detectors in space.

**Keywords:** silicon detectors; trackers; timing; LGAD; astroparticle detectors in space

## 1. Introduction

Cosmic Rays (CR) are messengers from the universe that, with the recent opportunity to operate precision particle physics detectors in space, stand as major probes to investigate astrophysical processes (with both Charged CR (CCR) [1,2] and photons at all wavelengths: radio [3,4], microwaves [5–8], IR and sub-mm [9–13], optical and UV [14–17], X-rays [18–20], $\gamma$-rays (GR) [20–22]) and also fundamental physics (Dark Matter [23–26], Gravitational Waves [27], Antimatter Asymmetry [28–30], Cosmology [31]), producing unique and complementary information to what is provided by experiments in laboratories at ground.

Most operating and planned space detectors for CCR and GR measurements require solid-state tracking systems based on Si-microstrip (SiMS) sensors. The feasibility of operating such detectors in space and their performances have been demonstrated by the successful operations of AMS-01 [32] and confirmed by the following missions (e.g., PAMELA [33], Fermi-LAT [22], AGILE [20], AMS-02 [1], DAMPE [34]).

In spectrometric experiments, such as PAMELA and AMS-02, tracking systems based on several layers of SiMS sensors are placed inside a magnetic field volume to accurately

measure the coordinate crossing of each particle to infer the trajectory curvature and consequently measure the particle rigidity. In calorimetric experiments, like Fermi-LAT, AGILE and DAMPE, in which the energy of the incoming particle is estimated using a calorimeter, tracking systems based on SiMS sensors are used to accurately measure the incoming particle direction. Moreover, for these latter experiments, a fraction of the tracker layers are interleaved with a high-density material in which GR can convert in a $e^{\pm}$ pair: the tracking system, in this case, is used to separately reconstruct the direction of the $e^{\pm}$ pair and, together with the energy measurement from the calorimeter, to reconstruct the four-momentum of the incoming GR. Depending on the physics target of the experiment, the SiMS signal is read out digitally to provide only coordinate information, or the additional $dE/dx$ signal can be read out to measure the particle charge to identify $Z > 1$ ions. Finally, the analog readout of the SiMS signals allows the position resolution to be improved using the charge sharing mechanism [35].

Future experiments aiming to reach higher energies and improved sensitivities will need to cover larger surfaces with Si detectors, with a substantial increase in the number of readout channels (e.g., e-ASTROGAM [36], AMEGO [37], PANGU [38], HERD [39], ALADInO [29], AMS-100 [30]). An increase in the total area of the Si detectors results in a direct increase in the number of electronics channels. The next generation of space detectors will face harder challenges in satisfying the power-budget availability with respect to current experiments (Table 1). Still, SiMS detectors are the most promising candidate solution to instrument such large areas while coping with the limitations on power consumption in space.

**Table 1.** Main parameters of operating and future Si-trackers in space [1,29,30,40–42]. In the table, the column "Strip-Length" provides the length or the range of lengths of SiMS "ladders" made of neighboring sensors connected in daisy-chain configuration.

| | Operating Missions | | | | | |
|---|---|---|---|---|---|---|
| | Mission Start | Si-Sensor Area | Strip-Length | Readout Channels | Readout Pitch | Spatial Resolution |
| Fermi-LAT | 2008 | $\sim 74\,m^2$ | 38 cm | $\sim 880 \times 10^3$ | 228 μm | $\sim 66\,μm$ |
| AMS-02 | 2011 | $\sim 7\,m^2$ | 29–62 cm | $\sim 200 \times 10^3$ | 110 μm | $\sim 7\,μm$ |
| DAMPE | 2015 | $\sim 7\,m^2$ | 38 cm | $\sim 70 \times 10^3$ | 242 μm | $\sim 40\,μm$ |
| | Future Missions | | | | | |
| | Planned Operations | Si-Sensor Area | Strip-Length | Readout Channels | Readout Pitch | Spatial Resolution |
| HERD | 2030 | $\sim 35\,m^2$ | 48–67 cm | $\sim 350 \times 10^3$ | $\sim 242\,μm$ | $\sim 40\,μm$ |
| ALADInO | 2050 | $\sim 80$–$100\,m^2$ | 19–67 cm | $\sim 2.5 \times 10^6$ | $\sim 100\,μm$ | $\sim 5\,μm$ |
| AMS-100 | 2050 | $\sim 180$–$200\,m^2$ | $\sim 100$ cm | $\sim 8 \times 10^6$ | $\sim 100\,μm$ | $\sim 5\,μm$ |

While the current SiMS detector technology already meets the minimum requirements for accurate position measurements in tracking systems and could be promptly equipped in the next-generation CCR and GR space detectors, possible additional improvements have the potential to enable new features and unprecedented accuracy in SiMS detectors, enabling de facto improved performances and, consequently, widening the physics reach of the whole space instrument.

Operating SiMS sensors with accurate timing capabilities in astroparticle detectors will provide breakthrough advances in the measurement of CRs in space. SiMS sensors could also have the ability to provide the time of particle crossing in addition to the 3D measurement of its crossing position for each measurement layer. The feature of accurate timing tracking ability (*4D tracking*), with a timing or velocity ($\beta$) value associated to each track/particle, has, in the space environment, interesting breakthrough applications, as described later in this document (cfr. Section 2). Additional novel experimental techniques can be further enabled if the spatial and energy deposit information of the particle crossing is integrated with the timing information, opening the possibility of precision single-hit

timing tracking, or *5D tracking*, with SiMS detectors in space. In the literature, the definition of *5D tracking* is not unique or unambiguous. Throughout the document, we will adopt a definition that integrates both definitions in [43,44]: with *5D tracking*, we refer to a 4D tracking where timing information is associated to each hit in the tracker, in a high-rate environment, and where each hit also has associated energy deposit information. The novel experimental strategies and advances created by this opportunity are the main subject of this document.

Although an unprecedented single-hit timing $\sim 130$ ps resolution, able to compete with those of scintillating devices, was achieved by the NA62 collaboration with standard Si-pixel sensors [45,46], this should be considered as a bound performance achievable with conventional planar Si technology. The geometrical layout and technology currently adopted for Si-sensors, in fact, are the limiting factors to obtain better performances for Si-pixel sensors and, most likely, to reach these resolutions for SiMS detectors [47,48]. New technological approaches and geometrical layout optimizations are consequently being investigated to enable more performant timing measurement abilities with Si-sensors. This could allow for comparable or even improved performances with respect to other timing devices, while keeping the mechanical properties and measurement abilities of solid-state devices.

Although there are few technological solutions available to allow for very performant time capabilities in Si-sensors [49], the Low Gain Avalanche Detectors (LGAD) is the most suitable, and a mature enough, candidate Si-sensor technology [47,48] to enable 5D tracking, simultaneously using very thin but efficient [50,51] SiMS detectors in space. The "3D sensor" technological approach is, for example, a possible feasible technology that may provide excellent timing resolutions [52], but it seems to not be suitable for large tracking areas (several m$^2$), with low power budget consumption (few or fraction of kW), as required for CR space measurement applications.

The LGAD technology integrates the features of standard Si sensors with an intrinsic gain layer typical of Avalanche Photodiode (APD) devices. Very thin LGAD detectors can consequently yield large enough signals to achieve timing resolutions down to 30–40 ps [48,53,54]. A more detailed technological discussion of LGAD sensors for timing applications is presented in Section 3. The maturity of the technology is confirmed by the fact that, as of today, LGADs produced from different vendors with different processes feature comparable performances. This makes this technology eligible for investigations of possible unconventional applications, such as those discussed in this document.

In this document, we mainly analyze the experimental advantages in the prospects of 5D tracking in astroparticle experiments, briefly describe possible technological solutions for its implementation, and finally comment on the technological path towards enabling 5D tracking in space.

## 2. Advantages with 5D Tracking in Astroparticle Experiments

Independently from the specific technology, the adoption of Si-tracking sensors with hit timing capabilities with a resolution of $\mathcal{O}(100\,\text{ps})$ will provide a breakthrough technology for tracking in space, enabling unprecedented solutions to future astroparticle experiments [55] such as:

1.  **identification of hits of back-scattered particles from calorimeters and improved track finding.** Future experiments based on deep calorimeters for the measurement of supra-TeV CCR will face the challenge of the loss in tracking efficiency at high energy due to the experimental noise introduced by the massive production of back-scattered secondary particles in the calorimeter, as already observed in large acceptance calorimetric experiments operating in space [56,57]. In standard Si-detectors, the hits coming from energy deposits by secondary-back-scattered particles cannot be separated from those of the primary particle. As a consequence, the efficiencies of hit clustering and particle tracking are affected. The relevance of this effect worsens with the number of back-scattered particles, and, ultimately, with the energy of

the primary particle. The additional measurement of the particle crossing time in Si-sensors provides the required information to separate primary from secondary hits, profiting from the fact that hits from back-scattered particles are produced with a delay with respect to the primary particle hits. Generally, timed-hits add additional coordinates in the phase space that can be exploited by track finding procedures to distinguish different tracks with much higher efficiency. This is one of the main reasons that the timing layers have been considered for the High Luminosity phase of the Large Hadron Collider [58,59], but it also opens up several opportunities for large-acceptance space-born CCR detectors, for which pile-up event suppression will become a challenge;

2. **overcome the occurrence of "ghost" hits in SiMS detectors.** Hits from back-scattered particles, detector pile-up, particle fragmentation, $\delta$-rays or pair-production, and noise, all contribute to the "ghost" hit problem [60] that strongly affects the track reconstruction performances in SiMS detectors, in which strips are arranged in perpendicular directions for each tracking plane. Peculiar strip geometries (e.g., stereo strips) or irregular readout pitch patterns can be used to mitigate this effect. However, the possibility of separating the tracks in time will be a powerful tool to overcome the issue without complicating the detector geometry;

3. **provide a Time of Flight (ToF) measurement that is complementary or alternative to that usually provided by a fast readout of plastic scintillators.** Hit timing measurements with resolutions $\sim$100 ps or less will enable the opportunity to perform ToF measurements with the SiMS tracking detector, with competitive performances compared to those of conventional ToF detectors made by plastic scintillators with fast photodetector readout. In CCR space-borne experiments based on magnetic spectrometers, the particle velocity measurement by a ToF detector is used to distinguish downward- from upward- going particles, which is crucial for separating matter from anti-matter in CCRs. The combination of velocity from ToF and momentum from tracker allows also for particle mass identification, which is used to measure the CCR isotopic composition and, possibly, to identify heavy anti-matter [1,29,30,32,33];

4. **improved e/p identification.** The presence of low-energy (i.e., $v \ll c$) back-scattered hadronic particles from a shower identifies the primary CR as hadron. Separating electrons, positrons and photons from the overwhelming background of protons that constitute the 90% of the CCR composition is a major requirement for most CR experiments. An innovative use of the accurate timing measurement in tracking detectors upstream of the calorimeter for this purpose was recently proposed [55] (Figure 1): the back-scattering of an electromagnetic shower is made of ultra-relativistic particles, even for very low energetic primaries. The detection of very delayed hits from slow back-scattered particles is a clear signature of an hadronic component in the shower, strongly suppressing the likelihood of an electromagnetic shower in the calorimeter.

These solutions are further enriched by the observation that, in general, when designing an apparatus, one should cope with the strong limitations in dimensions, weight and power consumption required by the launch and the operation in space. Equipping the experiment with a sub-detector with new measurement abilities from the same weight and dimensions represents a unique added value to the scientific mission.

Not all solutions 1 to 4 have similar advantages for CCR and for GR detectors. Similarly, the discussed solutions may result in different advantages for magnetic spectrometers and for calorimetric experiments. To mention a few cases, solution 3 does not apply to a detector developed for GR detection only based on the layout of current space missions. The same solution for a calorimetric-only detector, considering the typical hadron energy resolution achievable in space (30–40%), will not provide improvements in the mass measurement resolution, as the latter is dominated by the finite energy measurement resolution ($\frac{\sigma_M}{M} = \frac{\sigma_E}{E} \oplus \beta^2 \gamma^2 \frac{\sigma_\beta}{\beta}$). The reader should, however, value these considerations in view of the current technology that is being operated in space and for standard configurations of CCR and GR space detectors. Technological advances and novel ideas for future detectors may

enable an opportunity for relevant gains in performances by the application of 5D tracking, which is not currently amongst the most evident prospects described in this Section. Moreover, the possibility of operating 5D tracking detectors in space could possibly result in unprecedented novel layouts for future CCR and GR experiments designed around this measurement concept.

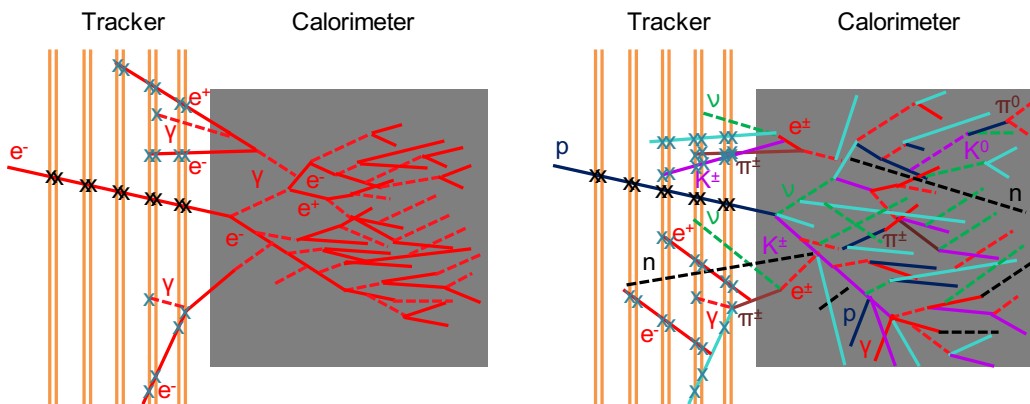

**Figure 1.** Representation of the secondary particle tracks in a tracking detector upstream of a calorimeter. Primary electrons (**left**) generate an electromagnetic shower in the calorimeter which may feature only ultra-relativistic $e^{\pm}$ and $\gamma$ backsplash secondaries in the upstream detectors. Interacting primary protons and nuclei (**right**) generate an hadronic shower (here sketched for display purpose only) in the calorimeter, which may feature a component of slow backsplash secondaries in the upstream detector.

Solutions 1 to 4 may require different levels of minimum hit time resolution performances to achieve the mentioned breakthrough advances. The minimum requirements strongly depend on the layout and scientific objectives of the whole instrument. Nonetheless, a target requirement of minimum $\mathcal{O}(100\,\mathrm{ps})$ hit timing resolution is a reasonable figure of merit that defines such technological target. In the next paragraph, some of the prospects described in this Section will be verified with a simulation of a demonstrator instrument with a $\mathcal{O}(100\,\mathrm{ps})$ hit measurement resolution baseline.

*Testing Prospects with Simulations*

A simple simulation was set up to verify the prospects for the advantages described in Section 2. The simulated detector is based on a typical layout of telescopic detectors with a tracking system upstream of a calorimeter. The tracker layout is based on that of the DAMPE SiMS tracker [41]. In the simulation, it is composed of 10 SiMS layers, each made of 300 μm thick sensors with 9.6 cm side squared area. Each sensor features 150 μm (50 μm) readout (implant) pitch with 640 total strips per sensor. A total of 64 sensors are arranged in an 8 × 8 chessboard geometry with strips running in the same direction to make up one layer. Four neighboring sensors, on both sides of each layer, are daisy-chained ("ladder") such that a single Front-End Electronics (FEE) channel reads out a 4-sensor-long strip. Pairs of layers with strips running in perpendicular directions are coupled in hodoscopic configuration with a distance of 2 mm over 5 planes. The distance between each plane and between the last plane and the calorimeter is 2 cm, which corresponds to a time of flight of ~65 ps for relativistic particles. The calorimeter is a 60 cm side cubic homogeneous Bismuth Germanate (BGO) monolithic volume, whose role in this study is limited to simulating the production of secondary back-scattered particles detected in the tracker. Figure 2 represents a sketched drawing of the simulated detector.

In this study, we have simulated the timing response for all readout sensors of the reference detector to verify the proof of working principle for 5D tracking in astroparticle experiments. Operations of similar detectors in space may require power mitigation techniques to cope with the limits imposed by the space mission environment, which

may impact the detector performances. Power mitigation techniques will be discussed in Section 3. Possible effects in the performances of 5D tracking depend on the specific implemented layout, and they will be studied in detail in future publications for a study-case space mission.

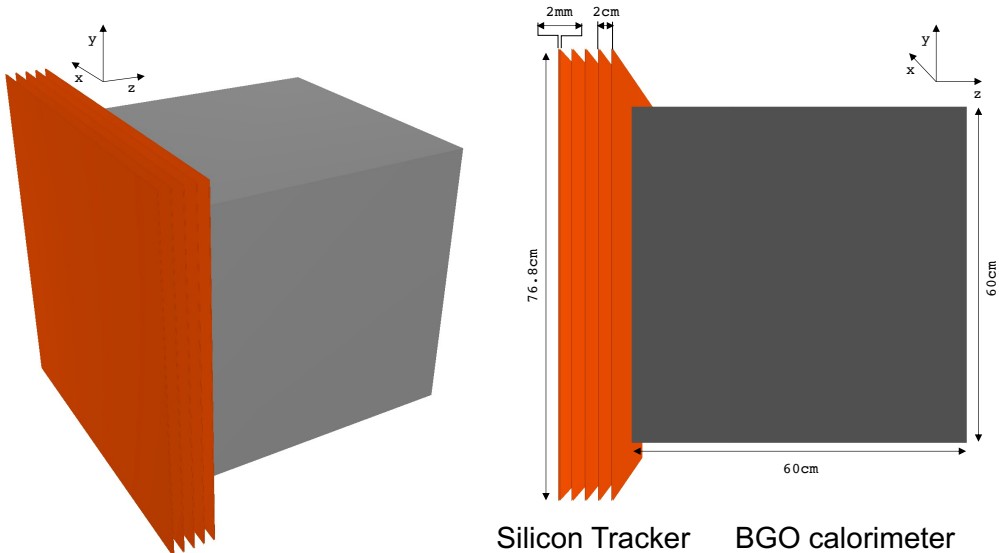

**Figure 2.** Layout of the simulated detector used in this work to demonstrate and quantify the prospects for timing measurements with Si-microstrip detectors in space.

The Generic Geant4 Simulation (GGS) software [61] suite was used to simulate the detector layout and the propagation and interactions of particles inside the detector materials. A thorough modeling of the sensor signal generation and shaping is beyond the scope of this work. The parametrization for signal generation and digitization used in this work is instead intended to provide an effective and fast simulation of the signal shapes and of the noise level, taking advantage of the experience gained with the AMS and DAMPE SiMS trackers [41,62] in order to obtain realistic position ($\sim$15 µm) and time ($\sim$100 ps) resolutions.

Electrons and protons were generated from a spot upstream of the tracker with enough beam divergence to illuminate the central sector of the tracker and the circle inscribed in the bottom face of the calorimeter with projected primary tracks. This beam geometry ensures that a large fraction of the showers are laterally contained in the calorimeter and minimizes the fraction of back-scattered particles outside of the tracker acceptance. This setup is representative of a test of the detector at a particle beam, but the considerations from these tests can also be applied for an isotropic illumination of the detector, since inclined particles in the acceptance and field of view of the whole instrument feature a larger time of flight through the tracker layers than those coming from the generation spot of the beam.

Figure 3 (left) shows the distributions of the true arrival time in the tracker sensors of primary 1 TeV protons and of the secondary particles generated by interactions of the primary protons with the detector materials, mainly with the calorimeter. The presence of hits generated by secondaries promptly produced with the upstream tracker materials before the interaction with the calorimeter are visible at low arrival times, but most of the secondary hits are dominated by back-scattered relativistic protons and $e^{\pm}$ with long tails beyond $\mu$s delays mostly from slow neutrons. Figure 3 (right) shows the inclusive distribution of time arrival measurements if the timing resolution of 100 ps rms from signal generation is applied, zoomed over a 2 ns time range from the first hit of the primary. The distribution of the true (MC truth) arrival times is superimposed to identify the different populations. The distribution of back-scattering hits is well separated from that of primary

hits. If the timing information is associated with each tracker hit, an upper bound selection can identify most back-scattering hits. Secondary hits from interactions with the upstream tracker cannot be resolved in this approach, but fragmentation events can be separately identified by correlations with large occupancies in the tracker layers. While the back-scattering hit identification ability strongly depends on the detector layout and on the timing resolution, this result clearly shows that, in principle, strong back-scattering hit suppression in the tracker can be achieved with related improvements in track-finding algorithms. This opportunity (1), together with the unambiguous and straightforward solving of "ghost" hits in the SiMS detector (2), will strongly improve the track finding and track reconstruction efficiencies in high occupancy or pile-up events.

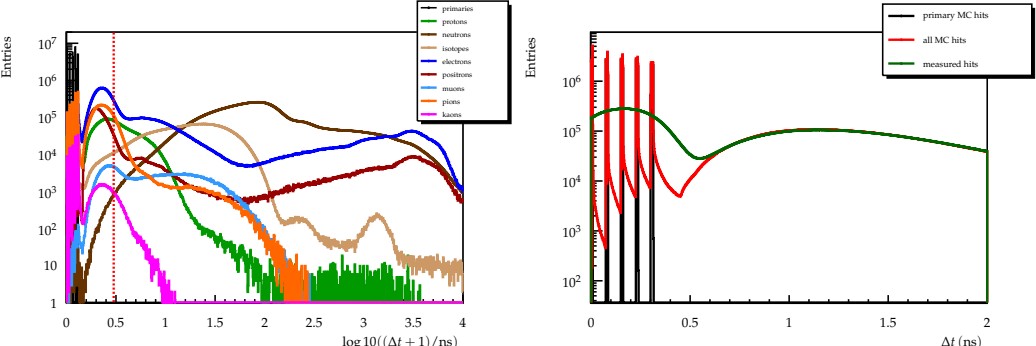

**Figure 3. (Left)**: Distributions of the true arrival time in the tracker sensors of primary 1 TeV protons (black) and of the secondary particles generated by interactions of the primary protons with the detector materials. Each entry represents the timing information associated to one hit in the tracker. A dashed vertical red line indicates a delay, from the first primary hit, of 2 ns, that is the time range in the figure on the right. **(Right)**: for the same events, the inclusive distribution of true arrival times (red) with the superimposed distribution of measurements assuming a timing resolution of $\sim$100 ps (green). The distributions are obtained out of $\sim$5 million generated events. In the distributions, we consider "hits" all the energy depositions in the sensitive volumes above a certain threshold ($\sim$10 keV, that represents the amount of ionization energy deposit resulting in a readout signal comparable to the typical FEE noise), also including energy depositions different from ionization.

Besides improvements strictly related to tracking, timing knowledge may provide information useful also for other applications, such as particle identification. The identification of $e^{\pm}$ CRs and their separation from the more abundant proton background is, for example, a typical figure of merit for astroparticle experiments. An electron/proton separation (e/p) of at least $\sim 10^5$ is required to achieve a precise measurement of the $e^{\pm}$ component in CRs. At high energies, e/p separation is provided by 3D shower topology imaging in calorimeters, by the yield of X-ray transition radiation in gaseous detectors and by the presence of slow neutrons in the shower components downstream of the calorimeter [63,64]. The investigation of the hit timing footprint from back-scattered secondaries in the tracker could provide additional, independent information to further improve the e/p separation abilities of the whole detector. To investigate this possibility, we have analyzed the difference in the arrival time of secondary back-scattered particles from proton- and electron-generated showers. Figure 4 shows the distribution of the arrival time of all hits in the tracker (left) and the same distribution limited to the latest hit in the event (right) for electrons and protons. Since a typical CCR analysis is performed in bins of energy in the calorimeter, for a fair comparison, the two species are compared only using events with similar deposited energy in the calorimeter: 700 GeV electrons and 1 TeV protons depositing 600–800 GeV in the calorimeter.

Both distributions show a clear difference between the two species, which could be explored in e/p separation algorithms. The distributions also confirm the naive idea depicted in Figure 1: proton events feature a longer tail of timing measurements due to slower secondaries. On average, proton and electron events feature 20 hits per events,

double what is expected by the signature of the primary particle only. The distribution for all tracker hits is dominated by a pronounced peak in the "prompt" back-scattering of secondary $e^{\pm}$ and $\gamma$ in the case of primary electrons and by the tail of slower back-scattering secondaries in the case of primary protons. The timing information of all hits per single event can be crunched in a single classifier by means, for example, of a multivariate algorithm [65,66] to maximize the effectiveness of e/p separation. On the contrary, the distribution of the latest tracker hits provides larger e/p separation abilities by itself: the peak in "prompt" back-scattering is, in fact, strongly suppressed for protons, because the latest hit is either produced by the primary particle in events with no back-scattering in the tracker or is delayed much beyond 1 ns with respect to the primary hits. Although dedicated studies are required to quantify the separation power depending on the energy and on the detector layout, these preliminary results provide a robust confirmation that hit timing measurements in tracking detectors can provide additional and independent information to enhance the e/p separation capabilities of systems based on Si-trackers and calorimeters (4.), providing information that is strongly independent from what is measured by other detectors used for hadron background suppression. The development of a classifier based on tracking timing information by means of a multivariate algorithm, the quantitative evaluation of the e/p separation performances achievable for specific layouts and the study of the level of correlations between the particle identification from the timing information and that usually obtained by other techniques, such as calorimeter shower shape topology analyses, will be the subjects of a forthcoming work.

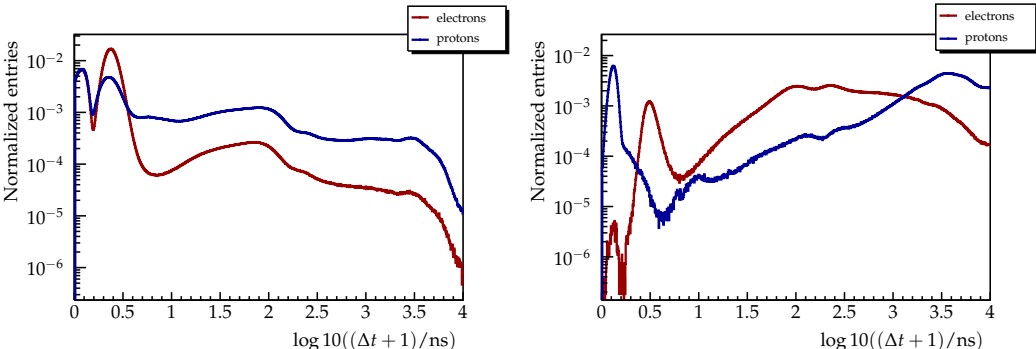

**Figure 4.** Inclusive distributions of time measurements assuming a timing resolution of $\sim$100 ps for 700 GeV electrons (red) and 1 TeV protons (blue), depositing 600 GeV to 800 GeV energy in the calorimeter. Each entry represents the timing information associated to one hit in the tracker. (**Left**): distribution for all tracker hits in the event. (**Right**): distribution for the latest hit in the event. We indicate, with "latest", the hit in the event which has the highest delay with respect to the first primary particle hit. The distributions are obtained out of $\sim$5 million generated events.

## 3. Technological Solutions

Although Si-pixel detectors are increasingly providing an excellent solution for solid-state tracking systems in a wide variety of applications, the most suitable candidate technology to instrument several m$^2$ of Si-tracker to be operated in space remains SiMS technology. In fact:

- the number of channels scales with the square root of the area to be covered (i.e., the side of the layers to be instrumented), to compare with pixels, for which it scales proportionally to the area. Clearly a factor 2 has to be taken into account to perform a fair comparison with a pixel detector measuring a pair of *X-Y* coordinates: SiMS can only measure just one direction so the number of strips, *n*, (i.e., side/pitch, $\frac{s}{p}$) is referred to a single coordinate measurement (for example, *X*). To obtain a *X-Y* coordinate measurement, 2*n* strips are needed;

- for pitch widths as those required for the aforementioned applications, the spatial resolution of SiMS with a readout pitch *p* is generally better than the corresponding

Si-pixel detector with a pixel size of $p \times p$, due to the insertion of floating implants (strips). Usually, in SiMS trackers, only a fraction of implanted strips out of a constant pattern is in fact read out. The remaining strips are floating and only contribute to the charge coupling between neighboring readout strips;

- besides coordinate measurements, SiMS also allow for a high-resolution charge ($|Z|$) measurement of the incident particle.

Simple computations [55] clearly show that, in space applications, the pixel geometry cannot be competitive, in terms of power consumption of the FEE, with respect to the microstrip one. The instrumentation of large area ($\mathcal{O}(10\,\text{m}^2)$) Si-trackers in space with pixel detectors requires a FEE with a power consumption of $\mathcal{O}(\text{nW})$/channel, a target that is realistically unattainable in the mid-term. Remarkable advances have been made in reducing the power consumption in Si-pixel sensors [67] especially in the case of Monolithic Active Pixel Sensors (MAPS) [68], but it still remains at the level of $\mathcal{O}(\mu\text{W})$/channel or fractions of it for MAPS. MAPS sensors with power consumption of a few tens of nW are conceivable [69], but not in the immediate future and most likely not with the desired features discussed later in the text. The limit for microstrip detectors covering the same areas loosens down to $\mathcal{O}(0.1\,\text{mW})$/channel, which is similar to the consumption of commercially available Application Specific Integrated Circuits (ASIC) for Si-detector readout in space (such as the $0.3\,\text{mW}$/channel of IDE1140 by IDEAS [70], formerly known as VA140 or VA64_hdr).

In addition to this, SiMS coupled to a High Dynamic Range (HDR) Charge Sensitive Amplifier (CSA) FEE can also measure the particle charge with a resolution of $\sim 30\%$ (10%) for a single measurement (combining more measurements) for high $Z$ ions ($Z \geq 10$) [71]. The effects of charge sharing and charge coupling also result in a better spatial resolution for SiMS than pixels with same readout pitch. For a given readout pitch, in fact, the spatial resolution for SiMS is usually better than the corresponding pixel equivalent due to the effects of charge sharing and charge coupling and the adoption of the floating strip. Roughly, for particles impinging perpendicularly to the tracking plane, the resolution is given by the pixel size (divided by $\sqrt{12}$) and, for SiMS, by the implant strip pitch (also divided by $\sqrt{12}$). For example, the AMS SiMS tracker has a readout (implant) pitch of 110 (27.5) µm, with a resulting resolution of $\sim 7$ µm [72]. For particles impinging perpendicularly, a pixel detector would require a readout pitch of $\sim 25$ µm, 5 times smaller than the SiMS readout pitch, to achieve a similar performance.

*Prospects towards 5D Tracking in Space with SiMS*

Implementing accurate and high-resolution timing measurement with SiMS, however, requires improving and enhancing the abilities of Si-detectors to cope with the challenging demands of next-generation astroparticle experiments. Although the requirements in terms of weight, volume and power consumption depend on the specifics of the mission and of the hosting space vector, the general expectation is that they will be much stricter than those applied to ground particle experiments.

A possible candidate technology, widely recognized to enable 5D tracking with SiMS detectors, is the LGAD technology. Improved timing resolution is, in fact, a compromise between reduced jitter (best in thick sensors or when the general signal is high) and high drift velocity uniformity and low Landau noise (best in thin sensors) [48]. The limited thickness of the Si-sensors, however, besides worsening the time walk, also reduces the signal yield and the Signal-To-Noise Ratio (SNR) of the detector. The LGAD layout, including an intrinsic "Gain" ($G$) layer, increases the Signal yield thanks to an avalanche mechanism and allows for recovery of the loss in signal yield and SNR in thin Si-sensors induced by the smaller amount of substrate. A 50 µm thick LGAD with $G = 10$ features the same signal yield of a 500 µm thick Si-detector. A 150 µm thick LGAD SiMS, for example, would have $\sim 4$ times the signal of the 400 µm thick sensors used for the Fermi-LAT tracker. As explained in Section 4, this can be interesting, for space experiments, either to reduce the material budget of the Si-tracker or to enable innovative experimental techniques [38].

Even if the LGAD technology can be implemented on both pixel and microstrip geometries [73], the requirements set by the expected high rates and high pile-up environments of the HL phase of the LHC have mostly uniquely driven the development of LGAD pixel sensors [43,58]. For these applications, in fact, the high power consumption of the pixel geometry is not prohibitive. LGAD strip geometries have been successfully built and operated [74–76] and, most likely, with a moderate R&D, it would be possibile to develop and build large Si wafers and the large sensor modules needed to instrument several m$^2$ of tracking devices.

Although the SiMS layout allows for the required position and timing resolution to be achieved, while mitigating the power consumption requirements, the power consumption for an FEE readout of charge and time for all the channels could exceed the limits set by operations in space. Analyzing the technology currently available in the market, the power consumption of FEE for timing is of the order of few mW/channel. As for example, we mention the PETIROC ASIC [77] which has been designed for Silicon PhotoMultiplier (SiPM) and not for SiMS. Nonetheless, SiPMs, feature an intrinsic gain from avalanches initiated by photoelectrons of at least $10^5$, which is of the same order of that for a LGAD SiMS sensor with $G = 10$, considering $12 \times 10^3$ electron-hole pairs produced in average by ionization in a thickness of 150 μm. In view of this, it can be considered a proper benchmark for comparison. The PETIROC ASIC features a power consumption of $\sim$6 mW/channel, which is a factor $\sim$10–20 higher than that needed just for the combined position and HDR charge measurement (cfr. above). Power consumption mitigation technologies could potentially enable low-consumption timing measurements with SiMS. However, power consumption mitigation techniques based on ad-hoc geometrical readout layouts could be promptly applied. We give a few possibilities below.

Simple power consumption mitigation layouts involve reading out groups of strips (e.g., 10 strips) with a unique FEE timing channel while keeping a separate strip FEE readout for charge/position measurement [55]. For example, assuming a timing FEE 10 times more consuming than the position/charge FEE, to limit the increase in the power consumption of just a factor 2, "grouping" $N = 10$ strip, only for the timing measurement, would be enough: each channel of the group (made of $N$ nearby strips or alternate strips with regular pitch jumps) is fed to a pre-amplifier and then to a fast shaper (tens of ns of peaking time, as opposed to few μs peaking time used for very low noise signal shaping in CSA) for the timing. The "OR" signal of the group of $N$ channels (generated, for example, after a discrimination step) is then digitized by a single Time-to-Digital Converter (TDC), as suggested in [55]. Using this approach for timing readout, charge/position measurements could be read out without modifications.

Depending on the chosen LGAD technology [74,78] and on the used FEE, different readout strategies can be adopted:

- use a single readout FE ASIC to read the strips with a CSA for position/charge, a fast shaper for the timing signal and generating a logical "OR" of the timing signals to feed a single TDC;
- use separate FE ASICs (e.g., IDE1140 and PETIROC) to read strips for position/charge measurement and strips (with larger pitch, for example, to reduce the number of channels) dedicated to timing. This requires a double-sided, such as inverse LGAD (iLGAD [74]), or a multi-layer, such as AC-coupled LGAD (AC-LGAD [79]) technology.

Different possibile strategies for the grouping of the SiMS are shown in Figure 5.

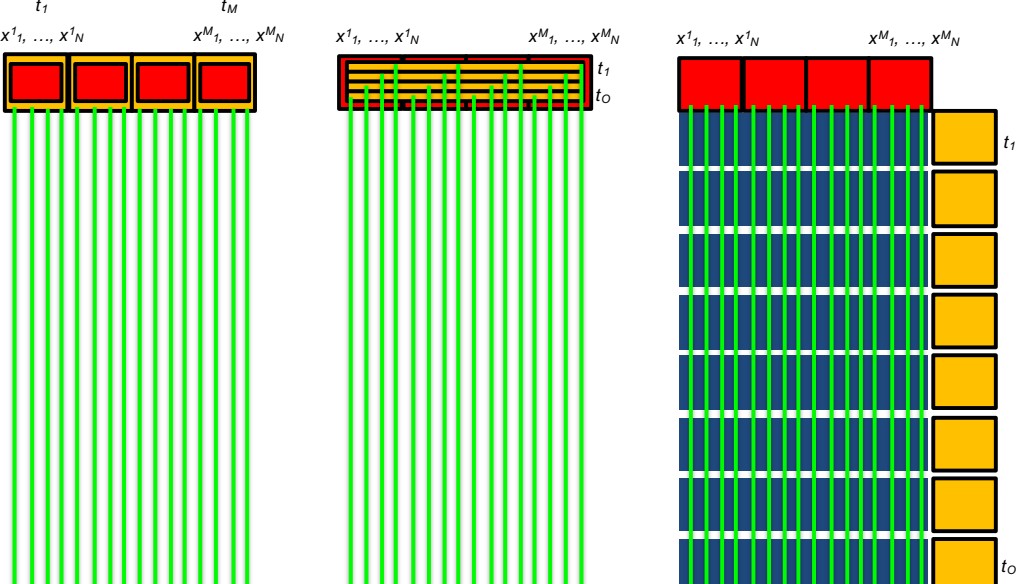

**Figure 5.** Conceptual sketch of three possibile arrangements of the position/charge (in red) and timing (with "grouping", in yellow) readout channels. (**Left**): the same strips are read for both position/charge and timing measurements. The readout FEE provides a single position/charge measurement per strip and a single time measurement per group ($N$ strips). This approach can be applied for any LGAD technology but requires a single FEE ASIC for both position/charge and time measurements. (**Center**): same as left, but the timing readout FEE $i^{th}$ channel reads out the parallel output of all $i^{th}$ strips for each group of $N$. This can be obtained with a double-side (e.g., in iLGAD technology) or multi-layer (e.g., in AC-LGAD technology) processing of the sensor or a custom, tailored, FEE ASICs. (**Right**): the strips for the position/charge and those for the timing measurement are orthogonal. This requires a double-side or multi-layer LGAD technology, but separate, FEE ASICs. Strips with larger width/pitch on the "timing" side are used to decrease the number of timing readout channels.

Other simple power mitigation layouts may be applied. In a "hybrid" approach, the timing measurement is enabled only in a fraction of the tracking layers, while for the remaining layers, only charge and position information are read out. The hybrid approach allows for large flexibility in defining the layers with timing information to maximize the detector performances while keeping the basic opportunities of 5D tracking.

Most importantly, further investigations are required to first identify for which the technology of SiMS LGAD sensors can be effectively compatible with fast timing readout FEE. Moreover, both grouping readout strategies and the hybrid layouts will have an impact on the detector performances, for example, on their tracking efficiency, timing measurement redundancy, velocity measurement and backsplash identification, which depend on the specifics of the layout itself. Thorough studies will be required to identify which solution or which combination is the most promising to maximize the detector performances while coping with the strict requirements of operations in space.

Besides power consumption, the radiation hardness of sensors must be carefully tested and verified to enable steady and long-term operations in space. The radiation resilience of LGAD sensors, a relatively recent technology, is currently being investigated in view of applications in detectors at hadron colliders. Recent results have shown that the performances of LGAD sensors exposed to irradiation tests remain acceptable up to fluencies of around $10^{15}$ $n_{eq}/cm^2$ [80,81]. These fluencies, which are those expected for applications of LGAD sensors planned for the High Luminosity LHC detectors, correspond to total ionizing doses larger than Mrad [82], which are, in turn, larger than those expected for years of spacecraft operations in most of the orbits. In space, single-event effects (SEEs) from heavy ions in the electronics circuits are the most dangerous radiation-induced

effects. Based on these preliminary investigations, we are confident that the finite radiation hardness of LGAD SiMS sensors will not represent a limitation for operations in space. Dedicated layout updates (also profiting from the experience on the ongoing R&D activities to improve the radiation hardness of LGAD sensors for collider physics detectors [80]), and irradiation campaigns, will nonetheless be required in the roadmap towards the space qualification of LGAD SiMS sensors to cope with and verify the possibility for steady long-term operations in the space environment.

## 4. Additional Opportunities from Operations of Thin Si-Microstrip Sensors

Besides the gain in timing resolution for LGAD described in Section 3, reducing the Si-sensor thickness will enable additional novel opportunities for next-generation large-area CR detectors and small-scale sub-GeV GR detectors. Operating thinner Si-sensors (150 μm instead of 300 μm used in AMS-02 and PAMELA) will reduce the material budget of tracking systems and will consequently improve the momentum resolution for spectrometers at low energies. In the spectrometer detectors recently operated in space, the Coulomb Multiple Scattering (MS) is, indeed, dominating the rigidity resolution up to several tens of GVs [83] (rigidity, $R$, is defined as momentum $p$ over charge $q$ ratio, $p/|q|$), while, at higher particle momenta, the finite spatial resolution of the tracking detector increasingly dominates the momentum resolution, leading to the momentum resolution parametrization $\sigma_p/p \propto p$. Although many experimental efforts in the technological development of tracking systems for spectrometers were conducted to improve the rigidity resolution at energies above 100 GV to search for new phenomena in this energy range [26,84–86], the momentum range below 10 GV is typically the only region where isotopic distinction is feasible [87] and where the momentum resolution dominates the mass measurement resolution ( $\frac{\sigma_M}{M} = \frac{\sigma_p}{p} \oplus \gamma^2 \frac{\sigma_\beta}{\beta}$ ). This is also a crucial region in the search for heavy primordial anti-matter signals [28,30,88]. Additionally, thin SiMS detectors enable the implementation of novel detection techniques also for GR instruments, especially for compact and sub-GeV detectors, providing unprecedented Point Spread Function (PSF) for MeV GR telescopes with a novel design based on fully active thin conversion layers with no passive converters [38]. However, decreasing the sensor thickness results in unavoidably lower SNR. Upgraded Si-sensor layouts, compared to the astroparticle detectors currently operated in space, such as LGAD sensors featuring intrinsic gain layers, are thus useful in coping with the loss of signal yield in thin sensors.

## 5. Conclusions

The operation of the current generation of large CCR detectors has opened a new era of precision particle physics in space. Large-area SiMS tracking detectors are typically primary subdetectors of CCR space experiments, and will probably continue to represent the most suitable solution for tracking devices in the near future. Nonetheless, novel technological improvements are needed to investigate the unexplored frontiers of CR in space with the next-generation astroparticle space-borne detectors with improved accuracy. In this document, we have discussed the possible advantages if, in addition to the well-established position and charge measurement, also precision single-hit timing measurement, are enabled in SiMS detectors. The advantages are many, and cover different applications, varying from improved track finding algorithms to e/p separation. A simple simulation of a typical layout of a telescopic detector with an upstream SiMS tracking detector, in combination with a downstream calorimeter, was set up to verify such advantages. Although the prospects strongly depend on the geometrical detector layout, these results show that a hit timing resolution of 100 ps, within reach of the technological developments described in this work, can enable unprecedented possibilities, such as backsplash hit identification and enhanced e/p separation. Adding such new features to the abilities of CCR experiments will surely enable breakthrough experimental advances for the measurement of particles in space.

Enabling 5D tracking in space demands a roadmap of technological development to assess the timing of performances in the envelope of the power constrains of space operations. Nonetheless, we point out that candidate technologies for these applications are already available, and we have identified the LGAD technology as the most suitable. Although the development of LGAD sensors has been mainly driven by its applications in solid-state pixel detectors for ground accelerator experiments, we have analyzed that the required R&D activities to develop and qualify LGAD SiMS sensors for space may be less demanding than what is required for applications in collider experiments. LGAD SiMS-based detectors could be developed to reach the technological maturity level in time to already equip the coming generation of space-borne CCR detectors with 5D tracking systems, de facto providing unprecedented experimental opportunities and improving the discovery potential of this research line.

**Author Contributions:** Writing—original draft preparation, M.D. and V.V.; writing—review and editing, V.F., M.G. and A.O.; project administration, M.D. and V.V.; supervision, M.D. and V.V.; funding acquisition, M.D.; software, M.D., V.F., F.F., L.M., A.S.; conceptualization, G.A., M.D., V.F., A.O., V.V.; validation, M.B., B.B., M.I.; formal analysis, E.C., F.D., G.S., L.T.; resources, B.B. All authors have read and agreed to the published version of the manuscript.

**Funding:** This research was partially funded by INFN grant number 19593/DTP (GRANT PER ATTIVITA' DI FORMAZIONE, PER SOSTENERE PROGETTI DI RICERCA DEI GIOVANI RICERCATORI VINCITORI DEI BANDI DI CONCORSO N. 18221/2016 E N. 18226/2016). The APC was funded by INFN grant number 19593/DTP.

**Institutional Review Board Statement:** Not applicable.

**Informed Consent Statement:** Not applicable.

**Data Availability Statement:** The code used in this work, which is in development and continuously updated, is publicly available and at https://github.com/bozzochet/DTP (accessed on 28 May 2021).

**Acknowledgments:** The authors acknowledge the contribution of the INFN colleague N. Cartiglia for the fruitful discussions on the possibility to have timing silicon sensors in Astroparticle physics detectors and for having encouraged the project since its conception. N. Cartiglia is also acknowledged, together with M. Boscardin and M. Centis Vignali from Fondazione Bruno Kessler (FBK), for their collaboration with the authors in trying to identify the more appropriate LGAD technology and to design a real large area LGAD SiMS sensor layout to be operated in space. The authors also want to acknowledge the INFN colleague N. Mori to be always available for discussions, bugfixes and feature requests on the GGS software. The authors would also like to acknowledge the fruitful discussions with the INFN colleagues M. Da Rocha Rolo and F. Cossio in the context of the "ASTRA" Front-End ASIC project.

**Conflicts of Interest:** The authors declare no conflict of interest. The funders had no role in the design of the study; in the collection, analyses, or interpretation of data; in the writing of the manuscript, or in the decision to publish the results.

## Abbreviations

The following abbreviations are used in this manuscript:

| | |
|---|---|
| APD | Avalanche Photodiode |
| ASIC | Application Specific Integrated Circuit |
| BGO | Bismuth Germanate |
| CR | Cosmic Ray |
| CCR | Charged Cosmic Ray |
| CSA | Charge Shape Amplifier |
| FE | Front End |
| FEE | Front End Electronics |
| G | Gain |
| GR | Gamma Ray |

| HDR | High Dynamic Range |
|---|---|
| IR | InfraRed |
| LGAD | Low Gain Avalanche Diode |
| iLGAD | Inverse LGAD |
| AC-LGAD | AC coupled LGAD |
| MAPS | Monolithic Active Pixel Sensors |
| MS | Multiple Scattering |
| PSF | Point Spread Function |
| R&D | Research and Development |
| SiMS | Silicon MicroStrip |
| SNR | Signal-to-Noise Ratio |
| TDC | Time-to-Digital Converter |
| ToF | Time of Flight |
| UV | UltraViolet |

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
