# Peer review of "Advantages and Requirements in Time Resolving Tracking for Astroparticle Experiments in Space"

_instruments, doi:10.3390/instruments5020020_

Round 1

Reviewer 1 Report

Dear Authors,

you suggest a technological solution for the next generation space detectors for cosmic rays based on the LGADs strip sensors coupled to dedicated readout ASICs. I find this suggestion worthwhile of further investigation, and while I still have reserves on the use of such a novel technology in an hostile environment, I think the idea should at least trigger some R&D work going into this direction. The first part of the paper explains the advantages of the proposed technology, in general terms and then more in detail with a GEANT simulation. The arguments and the simulation outputs are quite convincing, even if you did not perform a complete analysis (reconstruction) of the simulation results, so the full extent of the level of improvement (vs baseline SiMS) cannot be quantified.  
The second part of the paper focuses on the technology, in particular large LGAD (or iLGAD) silicon strip sensors. The considerations of this section are mostly reasonable, and supported by references, even if the section, being not supported by original experiments and/or devices stays very generic. 
Given the final statement of the article ("Realistically, the technological maturity level of LGAD SiMS could be reached in time to already equip the next generation"), I would also have liked a deeper  discussions on the problems that the current LGADs sensors show, and a path towards the development and large scale production of  LGAD SiMS (i.e. what has to be changed in the manufacturing process, what has to be tested and how).   

The paper is suggested for publication after revision.
A list of comments is attached.
Best Regards,
your reviewer.

The paper suggests a technological solution for the next generation space detectors for cosmic rays. The solution is based on the use of LGADs strip sensors coupled to dedicated readout ASICs able to record the hit time with a precision ~100ps, enabling a so called 4D tracking. The first part of the paper explains the advantages of the proposed technology, in general terms and then more in detail with a GEANT simulation. The arguments and the simulation outputs are quite convincing, even if sadly the authors did not perform a complete analysis (reconstruction) of the simulation results, so the full extent of the level of improvement (vs baseline SiMS) cannot be quantified.  
The second part of the paper focuses on the technology, in particular large LGAD (or iLGAD) silicon strip sensors. The considerations of this section are mostly reasonable, and supported by reference, even if the section, being not supported by original experiments and/or devices stays very general. 
Given the final statement of the article ("Realistically, the technological maturity
level of LGAD SiMS could be reached in time to already equip the next generation"), I would also have liked more discussions on the problems that the current LGADs sensors show, and a path towards the development and large scale production of  LGAD SiMS (i.e. what has to be changed in the manufacturing process, what has to be tested and how).   

The paper is suggested for publication after revision.
A list of comments is attached.
Best Regards,
your reviewer.

Title and text.
I think space should not be capitalized. 

Introduction 

It is not well explained in the introduction that such experiments are in reality quite different one from the other, and could fall in basically 2 categories identified by, for simplicity, AMS and Fermi (or GR/CCR). 
The paper would profit from a better separation of the two use cases and their respective requirements. As example, it is my opinion after reading the work that the proposed solution would be highly beneficial for AMS-like experiment and less so for Fermi-like ones. (I could be wrong, in any case this should not have been left to the reader).

L29-31
is a repetition of the previous sentence, can it maybe be merged?

L32
The continuity here might be lost. Maybe add "still" or "once again"?

Tab 1
Strip lenght (for the ladder) would be an interesting addition to the table.

In general it would be good to give in this introduction also an idea on how these detector systems are made, without necessarily looking at the reference. I recon that later in the paper you describe well the geometry of the simulated detector. Maybe you can reference that here. 

L44-50
I don't like these 2 sentences. On one hand, the "per-se" is kind of diminutive (it's already quite a breakthrough). On the other hand, they are just a peek into something you explain later, so they should be simpler to understand.

L47-48
you mean "high granularity of hits" or hits with high granularity information. The meaning of the former is not 100% clear.

L53
"with Si-pixel" -> "with standard Si-pixel"

L53
"hard limit" I would use another word, or better specify (e.g. "with conventional planar silicon technology"), since at L56 you suggest paths forwards.
On the other hand, 130ps is still extremely good, and probably good enough (you've used 100ps in the rest of the paper, would the result hold for 130ps?), so one could argue that this "hard limit" is not an issue.  

L65-69
strip sensor in 3D are possible, is just a matter of metal connections. Capacitance will be maybe to big (to meet the timing/noise requirement)? Anyway, I agree with the conclusion, 3D in space are unlikely.

L74-133
It would be useful if, for each of the different points, you could give an estimate of the time resolution needed. (e.g. 1 is likely much easier than 2-pp case)

L134 and following
In the simulation you're using the simple all 5D+calorimeter approach. The SiMS with LGADS and timing ROCS you propose are likely going to be more expensive, hard to operate, and power hungry than standard SiMS. HAve you investigated and hybrid approach, where you have N normal layers and 10-N timing layers (presumably at TOP and BOT, or all at BOT? 

L141
You should also mention that the 8 sensors in a ladder are readout, in parallel, with a single array of ROCs.

L155
'to obtain realistic" -> " needed to obtain realistic"

L162-164
Likely true, but as stated, unsubstantiated. Can you give an argument? or have you run some quick simulation to confirm? 

L172
"of∼100 ps" -> "of 100 ps rms"

L171
The long tails at >us must be (mostly neutron?) decays,  ions so slow  would have no energy to release, is it?    

Fig. 3L
Can you add a marker at 0.47 indicating the 2ns windows shown in Fig 3R?

Fig 4.
(blue) should be moved after protons, and a comma added.

L213
" slowest tracker hits" and note 4-> why not "latest"
L213-217 and Fig4R
if latest is later than 10ns, which seems very frequent, predictive power is quite reduced. What if you take not the latest but the latest 20% of tracks?

L210-122 and L218-222
Here the feeling is that you stopped just short of a proof. How difficult would be to run the suggested analysis and verify? It would make all more compelling.
Or maybe suggest it for a follow up?

L219
5D or 4D? Energy release per hit/plane has not come into play yet.

L231
add "due to the use of floating implant (strips) " (maybe move footnote 8 here?)
and "for large pitches as the ones here discussed."
L233
this is true for pixels as well (depending on readout)

L235 Space as adjective is for sure lowercase.

L241 Reference [64] is about timepix3, which is no MAPS.
MAPS with slow readout can reach extremely low level of power,
but don't have the features needed.
L245 In fact it is already the case for existing missions (e.g. Fermi).
In general, the paper would profit from a better separation of the considerations regarding existing vs speculative 

L250
Wouldn't  the floating strip configuration spoil partly the time resolution?
Is it compatible at all with LGADs? Unless you have the answers, some words of cautions are  needed here.
In theory, the same concept could be applied to a pixel system, even if the idea was not developed until now.

L282
"pixel LGAD Si-detectors" -> "pixels."

L284-286.
 6inch wafers are readily available, so I don't understand the mention :"large Si wafer". In any case, should be "wafers".
Also, you don't mention possible issues:
-)yield of big sensors in LGADs
-)stability (years of operation in space)
-)cost? is it comparable to a double side process or more expensive?

L305
"On the contrary" I fail to see the opposition here.
L306 "Conducted" -> done or performed 

L298-304
You're assuming that the timing performance are acceptable independently on the input capacitance. Are you confident that a 2mmx50cm superstrip with 10nf capacitance will deliver a 100ps time resolution? Reference to previous work? 

L304
"in short" even following the reference, it is not clear how electrically this is performed, unless a double sided sensor is used. 

L313 314
do you have references for the 2 acronyms?

L309-310
Again see my comment for L304, how is this performed?

L317-339
Here you cite some existing system  (AMS-02, PAMELA, DAMPE, AGILE320a nd Fermi-LAT)) but fail to point out that only the first 2 would profit from a reduction of material budget. The others have W converters in between Si layers.

L339
I would add, "if time resolution is desired". For position only, 150um standard sensors are still ok, with some improvement on the preamplifier.

L360
"assess the requirements of power budget limitation" The requirements will be given by the spacecraft. -> "to assess the timing performances in the envelope of the power constrains of space operation."

L367
"equip the next generation" -> "equip the coming generation" or some word that makes clearer you are talking of the "immediate" future.  

Author Response

Dear referee,

please find in attachment a pdf with our reply.

Beest regards,
The Authors

Reviewer 2 Report

It is a well elaborated work for astroparticle physics purposes. It certainly deserves publication. I would like to give some recommendations which the authors may be willing to follow now or in future work: 1. some practical numbers, if given, will widen the people being interested in. For example,      What is the energy threshold of the sensor? And how much it could be reduced?      What is the overall deadlayer of material in the entrance of the tracking detectors?       Even if it is not relevant for the immediate purpose of the set-up, it shows that it could be used        For other purposes in space starting with dark matter.       What is the total weight of the tracker system as it could be used also in balloon flights if not too heavy.   2. How large can become the overall gate width in time to eventually be sensitive to  slow speed        Exotica?  As such I mention the antiquark nuggets introduced by Zhitnitsky in 2003.   3. Intrinsic Gravitational selffocusing effects by the earth can enhance enormously DM fluxes on the opposite side of the earth.        This can open new pespectives. If taken into account appropriately.   4.  what is the total mass of silicon since it is also an active target for DM.

Author Response

(The authors gave the same response as above.)

Reviewer 3 Report

This paper studies advantages and requirements in time resolving tracking for astroparticle experiments. The document consists of five parts. In the first part, the authors introduce the proposed study together with relevant previous, current, and future works about the proposed field. The second part deals with the advantages of the 5D tracking in astroparticle experiments. Technological solutions are mentioned in the third part. In the fourth part, additional opportunities through operations of SiMS sensors are considered. In the fifth part, the conclusion is given.   

The paper is about the overview of the advantages and needs for large area Si detectors having a large number of electronic channels in the design of the next generation Space detectors. The power consumption issue in Space is also taken into account as an important point while developing the required experimental setups. The detector thought as the most promising candidate approach for these issues is Si-microstrip (SiMS) sensors which are analyzed in this paper. Besides the current usage of the SiMS tracking detectors taking data as 3D position information of particles, additional information provided by the detector such as time and energy deposit of each hit is proposed as a novel experimental strategy for the SiMS detectors employed in Space. Various detector technologies are compared to find the most suitable solution to the requirements including power consumption, weight, and size as well as experimental purposes such as precision physics studies in Space. The advantages of the 5D tracking method in astroparticle experiments are mentioned in the paper and verified by a simple simulation test setup study. The experimental method seems to be sound and correct. The results obtained by the simulation data prove clearly the main idea of the paper concerning the advantages of the 5D tracking. The main contribution of the paper is to find out the most suitable detector technology, which is the Low Gain Avalanche Detectors (LGAD) SiMS technology providing the requirements investigated in the paper. The best approach as a detector technology is found not only by the simulation setup results but also by many supportive and informative references, covering the whole idea of the paper. 

Minor revision:  the design of radiation-hardened components is also an important issue for the technologies sent to Space in addition to power consumption. The detector simulated as a Space technology in the paper may also need to be explained with a few sentences if it can be able to work under the Space radiation effects at various orbits like beyond the low Earth orbit. Therefore, I would expect from authors to add such type of information if applicable as a minor revision.

Some editorial comments that authors may take into account:

In line 44: … with timing …

In line 53: … hard limit …

In line 81: … the massive …

In line 91: … the delay …

In line 93: … with much higher …

In line 121: … that constitute 90% of … for most CR experiments …

In line 136: … upstream of a calorimeter …

In line 154: … taking advantage of …

In line 176: … is associated with …

Author Response

(The authors gave the same response as above.)
